# Imaging pain relief in osteoarthritis (IPRO): protocol of a double-blind randomised controlled mechanistic study assessing pain relief and prediction of duloxetine treatment outcome

Diane Reckziegel,[1,2,3] Helen Bailey,[1,2,3] William J Cottam,[1,2,3] Christopher R Tench,[4] Ravi P Mahajan,[1,5] David A Walsh,[1,6] Roger D Knaggs,[1,7] Dorothee P Auer[1,2,3]

For numbered affiliations see end of article.

**Correspondence to**
Professor Dorothee P Auer; dorothee.auer@nottingham.ac.uk

## ABSTRACT

**Introduction** Osteoarthritis (OA) pain is a major cause of long-term disability and chronic pain in the adult population. One in five patients does not receive satisfactory pain relief, which reflects the complexity of chronic pain and the current lack of understanding of mechanisms of chronic pain. Recently, duloxetine has demonstrated clinically relevant pain relief, but only in half of treated patients with OA. Here, the aim is to investigate the neural mechanisms of pain relief and neural signatures that may predict treatment response to duloxetine in chronic knee OA pain.

**Methods and analysis** This is an ongoing single-centre randomised placebo-controlled mechanistic study (2:1 (placebo) allocation), using a multimodal neuroimaging approach, together with psychophysiological (quantitative sensory testing), genetics and questionnaire assessments. Eighty-one subjects with chronic knee OA pain are planned to power for between-group comparisons (placebo, duloxetine responder and duloxetine non-responder). Participants have a baseline assessment and, following 6 weeks of duloxetine (30 mg for 2 weeks, then 60 mg for 4 weeks), a follow-up evaluation. Brain imaging is performed at 3T with blood-oxygen-level dependent functional MRI at rest and during pin-prick nociceptive stimulation for main outcome assessment; arterial spin labelling and structural imaging (T1-weighted) for secondary outcome assessment. Questionnaires evaluate pain, negative affect, quality of sleep and cognition.

**Ethics and dissemination** The study has been approved by the East Midlands, Nottingham and is being carried out under the principles of the Declaration of Helsinki (64th, 2013) and Good Clinical Practice standards. Results will be disseminated in peer-reviewed journals and at scientific conferences.

**Trial registration number** This trial is registered at ClinicalTrials.gov (NCT02208778). This work was supported by Arthritis Research UK (Grant 18769).

## Strengths and limitations of the study

► This study attempts to shed light on the little understood mechanisms of pain relief in chronic pain.
► By undertaking an innovative and integrative approach, which combines multimodal neuroimaging, quantitative sensory testing and psychometric questionnaire assessments, drug effects and predictors of treatment response to duloxetine are aimed to be identified.
► A crossover design would have ideally been employed. However, a parallel design was chosen to minimise dropout rates.

## BACKGROUND AND RATIONALE

Osteoarthritis (OA) is a major cause of long-term disability and chronic pain in the adult population. Knee and hip OA combined ranked 11th in a recent survey of global burden of disease, and symptomatic knee OA is several folds more common than hip OA.[1] Current treatment options consist of a combination of pharmacological and non-pharmacological alternatives.[2] Despite these options, around 20% of patients do not get satisfactory pain relief, even after undergoing joint replacement surgery.[3] This strongly supports the notion that chronic osteoarthritis pain has a strong central component.[4–6]

Application of advanced neuroimaging tools led to novel understanding of central mechanisms underpinning acute and chronic pain states.[7] Functional imaging in particular lends itself to the mechanistic study of analgesic treatment.[5 8–10] Neuroimaging may also

prove useful in guiding mechanism-based personalised therapy.[10–12]

To comprehensively describe brain activation states across regions and functional properties, the concept of so-called brain 'signatures' has emerged that can be derived from regional and network analysis of blood-oxygen-level dependent (BOLD) functional MRI data at rest or during controlled stimulation, or from cerebral blood flow maps based on arterial spin labelling (ASL) techniques.[13 14]

Duloxetine, a serotonin and norepinephrine reuptake inhibitor, has antidepressant and anxiolytic properties and is currently also used to treat chronic pain conditions. However, the precise mechanisms by which this drug acts to relieve pain are unclear and only around 50% of patients with OA have sufficient pain relief after duloxetine intake[15]; currently there is no way to predict responders.[15 16] In healthy participants, duloxetine was found to attenuate activity in affect processing areas (amygdala, thalamus, insula and anterior cingulate cortex) while simultaneously increasing functional coupling between the amygdala and the anterior insula during an emotional face matching task.[17] Duloxetine in major depression augmented connectivity in the anterior default mode network,[18] while reduced connectivity within the subgenual cingulate was predictive of clinical antidepressant response.[18]

The aim of this study is to investigate the neural mechanisms of pain relief in chronic OA pain following treatment with duloxetine and to establish multimodal brain signatures to predict treatment response. We will use a unique multimodal MRI design to identify key mechanisms and predictors of duloxetine-induced pain relief in chronic OA pain. This will include brain resting state functional connectivity, ASL, nociceptive functional (f)MRI, structural T1-weighted brain MRI, proton spectroscopy, quantitative sensory testing (QST) and questionnaire assessments.

We hypothesise that responding and non-responding patients with OA have different limbic signatures, differences in negative affect and sensory pain phenotypes. Additionally, we hypothesise that duloxetine induces functional brain changes in emotional regulatory networks.

## PRIMARY OBJECTIVES

The objective of this study is to identify a functional change in brain signature indexing the analgesic mechanism of duloxetine and to identify brain imaging and psychophysical markers that predict the response to duloxetine treatment.

## MAIN HYPOTHESES

► Analgesic response to duloxetine treatment can be predicted using a range of baseline brain imaging markers and QST.

► Analgesic response to duloxetine is mediated by modulation of neural networks underpinning emotional control.
► Duloxetine-induced changes in brain activation differ between responders and non-responders.

## DESIGN AND METHODS

This is a single-centre, double-blind, placebo-controlled mechanistic study being conducted at the University of Nottingham (UK).

### Inclusion criteria

To participate in this study, individuals must have chronic knee pain with radiographically defined OA changes (Kellgren Lawrence grade 2 or higher), be able to consent to participation and be aged 35 or older. They must also be free of major medical, neurological and psychiatric comorbidities known to affect neural processing of pain.

### Exclusion criteria

Potential candidates meeting any of the criteria listed below are excluded from participation.

General:
► Significant medical conditions, including uncontrolled hypertension, insulin-dependent diabetes, dementia, Parkinson's disease, epilepsy, multiple sclerosis, active cancer, major depressive disorder, mania, bipolar and schizophrenia. This is judged by a qualified physician based on medical records.
► Refusal by candidate to GP being informed.
► Planned total knee replacement.

MRI related:
► Intraocular metallic foreign bodies.
► Intracranial aneurysm clips.
► Cardiac pacemakers and defibrillators.
► Cochlear implants.
► Significant head tremor.
► Potential metal foreign bodies due to previous accidents.
► Breastfeeding or pregnancy, confirmed by pregnancy test.
► Unfitness for the MRI scanner, according to the judgement of medically qualified personnel, either on the research team or the patient's clinical team (eg, due to back pain, claustrophobia, acute sickness, etc). This includes patients with signs of impaired temperature regulation such as an extremely high fever.
► Patients with large tattoos, specifically in the head, neck or shoulder region.
► Metallic agents embedded within the body (ie, Shrapnel, surgical pins).

Duloxetine related:
► Uncontrolled narrow-angle glaucoma.
► Recent usage of monoamine oxidase inhibitor or Mellaril (thioridazine).
► Taking St John's Wort (Hypericum perforatum).

- ► On fluvoxamine, ciprofloxacin or enoxacin.
- ► Taking other medicines containing duloxetine.
- ► Liver disease.
- ► Renal impairment.
- ► Currently on antidepressant treatment, including treatment for pain with tricyclic agents such as amitriptyline.
- ► Taking tramadol.
- ► Known hypersensitivity, allergy or intolerance to one of duloxetine's components.
- ► Unwillingness to take caution in relation to use of other centrally active substances such as alcohol and sedative drugs.
- ► Current treatment with potent inhibitors of CYP1A2 like fluvoxamine.
- ► History of seizures or any drug lowering the seizure threshold.

### Randomisation and blinding

Participants are randomised between duloxetine (n=54) and placebo (n=27). The allocation ratio is 2:1; given that around half of the population undergoing active treatment is expected to have satisfactory pain relief, this should result in approximately equal numbers between groups of placebo intakes, responders and non-responders. A block randomisation, with block sizes of 9 and 2, is utilised and was generated by the Clinical Trials Pharmacy (CTP) at Nottingham University Hospitals (NHS Trust) using a pseudo-random algorithm at http://www.randomization.com. The allocation list is kept by the CTP in a password-protected file. Investigators may identify the treatment for each participant via request to the CTP; however, during treatment, this breaking the code procedure may only be done in the event of a medical emergency when treatment is dependent on knowledge of the actual drug received. In case the treatment code for a participant is broken or other actions, such as recording date or reasons for breaking the blind, discontinuation of

trial treatment must be reported to the sponsor (University of Nottingham).

Participants are assigned, by the investigators, a unique identification number (sequentially numbered) and attend the university twice, for both a baseline and follow-up visits. Treatment consists of 6 weeks of taking either duloxetine capsules or placebo equivalent, followed by 2 weeks of dose reduction for those taking duloxetine to minimise drug withdrawal effects (figure 1). To minimise potential side effects, participants take one tablet (duloxetine 30 mg) daily during 2 weeks prior to switching to the target dose of 60 mg/day. Both researchers and participants are blinded to the treatment allocation until data acquisition is complete. At the end of the follow-up visit, individual allocation is disclosed to the participant only by one of the CTP staff members who are not involved in the any data analysis or research procedures. Data analysts will be blinded to treatment allocation.

Individual participation in the study is for around 8 weeks, starting with signing informed consent and ending with the conclusion of the drug reduction phase.

### Experimental testing

The experimental testing is primarily based on multimodal neuroimaging and QST, in conjunction with questionnaire assessment and genetics. This should provide a broad evaluation of the multifaceted nature of pain, assessing both central and peripheral mechanisms, as well as psychological and cognitive traits, among others, which should allow pinpointing precise phenotyping of patients into distinct groups. Time points and types of data being collected are summarised in figure 2 and details of questionnaires are given below.

Questionnaires:
- ► PainDetect (baseline)—a screening questionnaire to identify neuropathic components of pain.[19]
- ► Pain Catastrophizing Scale (PCS) (baseline)—a 13-item scale to measure pain catastrophising, which

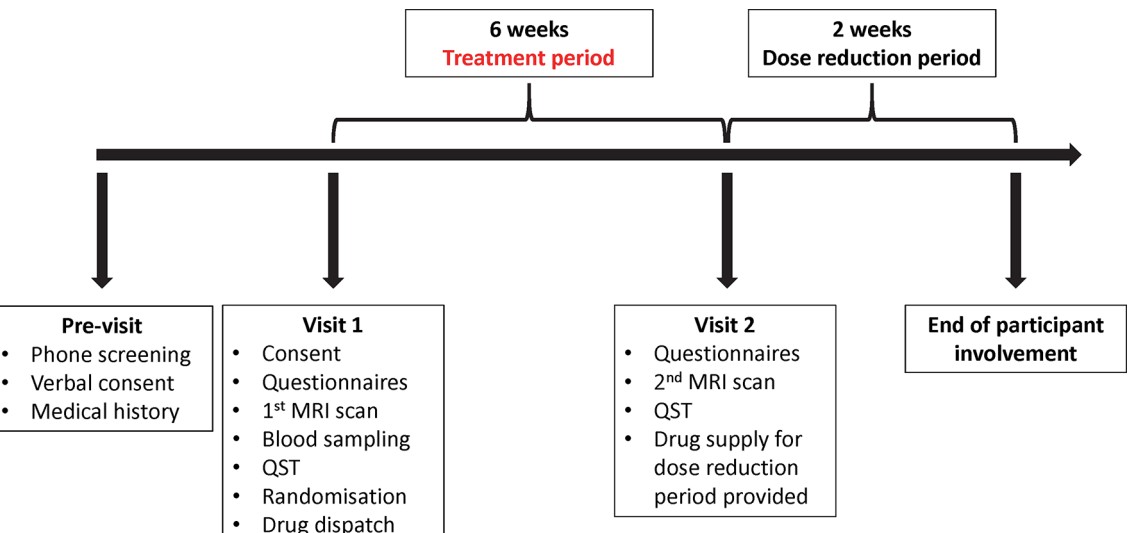

**Figure 1** Participation timeline and general characteristics of each study visit. QST, quantitative sensory testing.

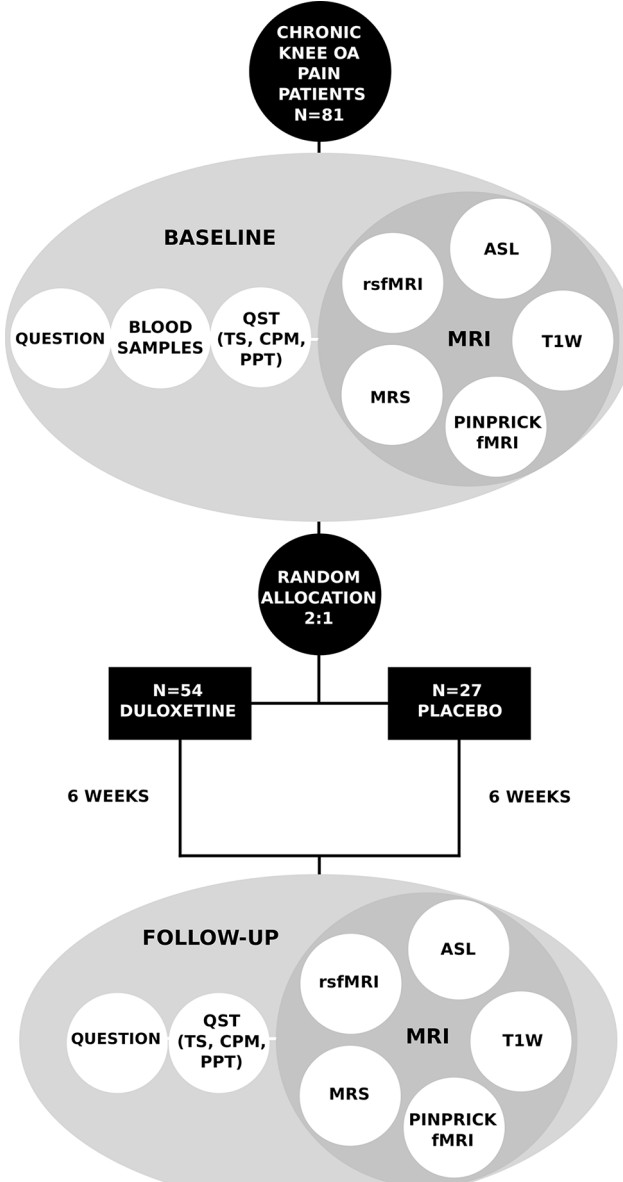

**Figure 2** Details of experimental testing and randomisation of participants. Volume of interest for magnetic resonance spectroscopy is located in the midanterior cingulate cortex. ASL, arterial spin labelling; CPM, conditioned pain modulation; fMRI, functional MRI; OA, osteoarthritis; PPT, pressure pain thresholds; QST, quantitative sensory testing; Question, questionnaires; rsfMRI, resting state fMRI; T1w, T1-weighted MRI; TS, temporal summation.

can be split into three subscales named rumination, magnification and helplessness.[20]
► State-Trait Anxiety Inventory (STAI) (baseline)—a psychological inventory assessing anxiety. It contains 40 items and can be split into two subscales being state anxiety and trait anxiety.[21]
► Beck Depression Inventory II (BDI) (baseline)—a 21-item multiple-choice self-report scale for measuring the severity of depression.[22]
► The intermittent and constant pain score (ICOAP) (baseline and follow-up)—an 11-question scale to

characterise pain in individuals with osteoarthritis pain. It gives two subscales relating to intermittent and constant experience of pain.[23]
► The Pain Sleep Questionnaire (baseline)—a five-item tool to assess the impact of pain on sleep.[24]
► Brief Pain Inventory (BPI) (daily)—a nine-item questionnaire for measuring the severity of pain and the impact of this pain on daily functioning. The severity subscale is used to define the response definition (below).[25]
► Mini-Cog (baseline)—contains a three-item recall test and a clock drawing test to screen cognitive impairment.[26]
► Educational level (baseline).
► Questions on belief (baseline)—consist of four questions to assess an individual's expectations on treatment and medical care.[27]

Participant general details:
► Age.
► Sex.
► Pain intensity (measured on a numeric rating scale ranging from 0 to 100, where 0 represents no pain at all and 100 stands for excruciating pain).
► Pain duration.
► Analgesia intake (last 24 hours).
► Caffeine intake (last 24 hours).
► Average daily caffeine intake.
► Average weekly alcohol intake.
► Medication.

Other:
► 10 mL blood samples.
► Analgesia diary (daily).

### Response definition
Response to duloxetine is defined as a 50% reduction in the severity of ongoing pain from baseline to end of treatment, as measured by the BPI severity subscale. This threshold was chosen based on previous randomised controlled trials of duloxetine in osteoarthritis.

### Primary outcomes
#### Treatment mechanism
1. Change in the regional response of BOLD contrast to pinprick nociceptive stimulation after duloxetine treatment (MRI2) versus baseline (MRI1) in treatment responder group.
2. Functional connectivity change from baseline (MRI1) to post duloxetine treatment (MRI2) using network metrics of BOLD fMRI under resting condition in treatment responder group.

#### Response prediction
3. Differences in nociceptive BOLD response and network function at baseline between responders and non-responders.

## Secondary outcomes

1. Difference in the regional response of BOLD contrast to pinprick nociceptive stimulation after duloxetine treatment (MRI2) between responder and non-responder groups.
2. Functional connectivity differences post duloxetine treatment (MRI2) using network metrics of BOLD fMRI under resting condition between treatment responder and non-responder groups.
3. Correlation between baseline CPM and TS with brain activity and connectivity changes from baseline to post 6 weeks of duloxetine treatment (rsfMRI and pinprick fMRI).
4. Group differences in brain activity (pinprick fMRI and rsfMRI) and structure (T1-weighted brain MRI) in pain processing, limbic and modulatory pathways changes from baseline to following 6 weeks of duloxetine treatment, in comparison to placebo.
5. Identification of baseline QST (CPM, TS, PPT) and questionnaire parameters (BDI, STAI, PCS, ICOAP) that predict response to duloxetine.
6. Multivariate treatment response prediction model.

### Other outcomes

Exploratory outcomes include, but are not limited to, determination of gene variations that can be linked with duloxetine treatment response and gamma-aminobutyric acid (GABA) modulation according to changes in pain severity. Imaging data from the baseline study may be pooled with other studies undertaken in the Nottingham Arthritis Research Pain Centre ARUKPC if the participants consent to this.

### Data analysis

Clinical, QST and psychometric data will be compared between groups and conditions using the latest versions of statistical and image analysis software including SPSS (version 21.0).

Brain imaging data will be analysed using the latest versions of several established toolboxes: FSL (FMRIB Software Library),[28] PyMVPA (Multivariate Pattern Analysis),[29] Statistical Parametric Mapping V.8 (London, UK), REST (Resting-State fMRI Data Analysis Toolkit), Brain Connectivity Toolbox and GRETNA (Graph Theoretical Network Analysis)[30–32] and LCModel (Linear Combination Model) for MRS data.[33] All data will undergo established quality control and standard recommended preprocessing. Preprocessing steps for functional data include motion correction, slice-timing correction using Fourier-space time-series phase-shifting, non-brain tissue removal, spatial coregistration to T1-weighted image and normalisation to the MNI anatomical standard space. Spatial smoothing will be carried out using an isotropic Gaussian kernel and a high-pass temporal filtering will be applied. Next, for task-related fMRI, statistical maps will be generated for contrasts of interest, such as to identify brain activity evoked by pinprick in each individual scan using first-level general linear model.

Z-scores will be extracted from regions of interest of pain processing nodes defined from meta-analysis,[7] emotion regulatory, default mode and salience network including anterior cingulate cortex, anterior insula, amygdala, brain stem and thalamus for both task-related and resting functional maps. Individual functional connectivity maps will be also generated. These will be used on higher level analysis which will include, among others: (1) Paired t-test to evaluate treatment effects; (2) two-sample t-test to compare groups (responders and non-responders) at baseline; (3) correlation analysis to determine associations with psychometric and QST measurements.

### Sample size estimation

The sample size calculation was performed according to the guidelines stated by Lenth and colleagues.[34] Estimates of change in functional activation were deduced from a duloxetine intervention study (effect size was calculated based on p values provided for the ACC).[17] Using the Java applets,[35] it was estimated that a one-sample, one-tailed, t-test would have 80% power at alpha of <0.005 (to allow for 10 multiple independent tests) with approximately 19 participants per group: responders to duloxetine treatment, non-responders and placebo groups.

Based on previous experience of non-usable imaging data due to patient and/or technical factors in 10% of cases, a sample size of >22 participants per group was necessary. A further allowance for 20% dropout gives 27 participants in each group, resulting in a recruitment target of 81 participants. The study is not be powered to assess clinical efficacy.

### Recruitment

Identification of potential participants is done by contacting people who have taken part in previous studies within the ARUKPC. Recruitment through primary care and posters is also being used. Candidates are prescreened during phone call following expression of interest. Medical notes are verified for eligibility following oral consent. Suitable candidates are then booked in for the first visit when written informed consent is obtained from all individuals taking part in the study by one of the investigators before any interventions related to the study will take place. The procedure consists of the following: the investigator explains the details of the study and provides participant information sheet and drug information leaflet, ensuring that the participant has sufficient time to consider participating or not. The investigator then answers any questions that the participant may have concerning study participation before signing the consent form.

### Concomitant medication

Participants are instructed not to change their regular pain treatment while taking part in the study. Rescue pain medication, taken on an 'as needed basis', is accepted during the trial period and its use is documented in a daily pain diary.

## Compliance

Compliance is assessed in the first instance by their attendance at all scheduled research visits. Participants who choose to discontinue are asked to inform the investigators on their decision. Participants are contacted via telephone while undertaking the study medication to promote participant retention and completion of follow-up. During the second visit, participants are asked whether they took the drug according to the protocol.

Should the participant be unwilling to reschedule their appointment, data collected up until that point will be carried forward to the analysis stages and they will be withdrawn from the investigation. If a participant is unable to return for their second visit, follow-up questionnaires will be obtained by post.

## Criteria for withdrawal

Participation in this study is voluntary and participants are free to withdraw from the trial at any time, without giving any reason. However they must seek advice from the researchers before doing so as discontinuing duloxetine treatment suddenly may cause drug withdrawal effects. Involvement may be also discontinued by the investigators in case of any significant adverse effect of duloxetine/placebo treatment or when subjects require major changes to their ongoing pain management during the course of the study.

## Adverse events and data safety monitoring

All observed or reported adverse events (AEs) are recorded by the investigators. No AEs are expected in relation to the MRI scanning. Potential AEs are anticipated to be only related to drug intake. All adverse events are recorded and monitored until resolution, stabilisation or until it has been shown that the study intervention is not the cause. Any serious adverse event is recorded and reported to the research ethics committee as part of the annual reports.

## Handling and storage of data and documents

All study staff and investigators endeavour to protect the rights of the trial participants to privacy and informed consent, and will adhere to the Data Protection Act, 1998. Personal data is handled confidentially. Imaging, sample and questionnaire data are anonymised by assigning a unique participant identification code, containing the abbreviation of the study's name followed by a sequence number. The link between study identification code and individual's personal details is kept on the consent form and MR screening form.

Computer-held data are kept securely and password protected. All data are stored on a secure dedicated server. Access to data is given to the investigators and is restricted by user identifiers and passwords. Copies of the raw imaging data are also stored on CD/DVD in a locked room in the Radiological Sciences premises as back-up

data. Either double data entry or checks of 40% of data entry are to be performed to promote data quality.

Direct access is permitted when required to all source documents and other study documentation for example, signed consent forms, for the purpose of study monitoring and audit and other lawful regulatory inspection by the chief investigator, sponsor's designee and inspection by relevant regulatory authorities.

The investigators have full access to the dataset without limiting contractual obligations.

## Monitoring trial conduct

Trial conduct is subject to systems audit of the Trial Master File for inclusion of essential documents, permissions to conduct the trial, trial delegation log, curriculum vitae of trial staff and training received, local document control procedures, consent procedures and recruitment logs, adherence to procedures defined in the protocol (eg, inclusion/exclusion criteria, correct randomisation, timeliness of visits), adverse event recording and reporting, drug accountability, pharmacy records and equipment calibration logs.

The trial coordinator or a nominated designee of the sponsor shall carry out a site systems audit according to local requirements.

**Author affiliations**

[1]Arthritis Research UK Pain Centre, University of Nottingham, Nottingham, UK
[2]Sir Peter Mansfield Imaging Centre, University of Nottingham, Nottingham, UK
[3]Division of Clinical Neuroscience, Radiological Sciences, University of Nottingham, Nottingham, UK
[4]Division of Clinical Neuroscience, Clinical Neurology, University of Nottingham, Nottingham, UK
[5]Division of Clinical Neuroscience, Anaesthesia and Critical Care, University of Nottingham, Nottingham, UK
[6]Division of Rheumatology, Orthopaedics and Dermatology, University of Nottingham, Nottingham, UK
[7]School of Pharmacy, University of Nottingham, Nottingham, UK

**Contributors** DPA and DAW designed the study and DR, DPA, RDK, CRT and RPM developed the research protocol. DR leads the study, is responsible for data acquisition and will conduct image and statistical analyses. CRT provides statistical consultation. DR, HB and WC undertake recruitment and data collection. DR and DPA wrote the manuscript, which was edited by CRT and RDK. All authors approved the final manuscript.

**Competing interests** None declared.

**Patient consent** Detail has been removed from this case description/these case descriptions to ensure anonymity. The editors and reviewers have seen the detailed information available and are satisfied that the information backs up the case the authors are making.

**Ethics approval** East Midlands—Nottingham 2 research ethics committee.

**Provenance and peer review** Not commissioned; externally peer reviewed.

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
