## [Reviewer comments · BMJ Open]

ARTICLE DETAILS

TITLE (PROVISIONAL)	Imaging Pain Relief in Osteoarthritis (IPRO): protocol of a double blind randomised controlled mechanistic study assessing pain relief and prediction of duloxetine treatment outcome
AUTHORS	Reckziegel, Diane; Bailey, Helen; Cottam, William; Tench, Christopher; Mahajan, Ravi; Walsh, David; Knaggs, Roger; Auer, Dorothee

VERSION 1 - REVIEW

REVIEWER	Leticia Deveza Rheumatology Department, Royal North Shore Hospital and Institute of Bone and Joint Research, Kolling Institute, University of Sydney, Sydney, New South Wales, Australia.
REVIEW RETURNED	27-Sep-2016

GENERAL COMMENTS	This trial aims to study the neural mechanisms associated with pain relief obtained with duloxetine and identify predictors of treatment response in patients with symptomatic knee OA. This is a very interesting study and question, specially in view of the heterogeneity of the population of knee OA patients. Following a few comments/suggesting to the authors regarding the protocol paper: 1) Abstract: It is not clear to me what will be used to predict response ("neural signatures"). If it is the QST and questionnaires in addition to the neuroimaging, please make this clear in the abstract. Also, I suggest adding that this is brain imaging to avoid any confusion.2) Page 3, line 36. "physiotherapy" is quite vague. As you said exercises before, I suggest taking this out or replace by the specific physio intervention.3) Page 6, line 9. Again, I suggest adding that this is brain MRI.4) Inclusion criteria: page 7: Any radiographic OA change will be enough for inclusion? (eg KLG 1-4 or KLG 2-4; or based on another method). Please be more specific on this radiographic criteria.5) Significant medical conditions: please be more specific and state who will judge this.6) Experimental testing, page 10, line 41: Clinical examination of the knee is mentioned here but no further details are provided. Are you using this as a potential predictor of response? And what is being assessed? Please clarify.7) Page 11, line 42: Please provide the full name of the "ICOAP" questionnaire before the acronym. I also suggest describing what each of these questionnaires is intended to measure and how they are scored.8) Page 12, line 20: "pain intensity" - measured how? (eg. VAS?)9) Response definition, page 12: Not explained how 50% in pain improvement will be measured and the rationale to choose this
---

	threshold. Also, there is no minimum pain requirement for the trial? For example, someone may have the pain scores reduced from 2 to 1 (if VAS 0-10), is it enough to be considered response? Please clarify. 10) Is there any requirement for the participants to avoid other pain medications before the assessments? (eg NSAIDs) This could minimize confounding when assessing the images. 11) Outcomes, page 13. Please explain how each one of these will be measured. I suggest linking the MRI techniques that was described before to each of these outcomes (and adding if rest or pin-prick for all of them). Please also explain the method that will be used to define change (if it is quantitative, semi-quantitative, etc). How is it defined? 12) Data analysis: Will the analysis follow the ITT principle or per protocol? Also, please include the complete statistical methods that is planned to be used for the analysis (for aims 1 and 2) 13) Sample size calculation: Page 14, Line 34: Please include here which parameters were used to calculate the sample size (that were "deduced from reference 20") so it is possible to judge if the sample size is adequate. 14) Will adherence to the treatment be monitored throughout the study? And how?
--	--

REVIEWER	Steven P. Cohen Johns Hopkins, U.S.A.
REVIEW RETURNED	05-Oct-2016

GENERAL COMMENTS	The authors are publishing the protocol of a study designed to determine mechanisms of analgesic response to duloxetine. 1. The introduction is very long. It would be too long for a manuscript in BMJ, and it's too long for a protocol. 2. If the authors hypothesize (page 6) that brain activation changes induced by duloxetine overlap with antidepressant mechanisms, why did they exclude people with depression (I understand that people with depression might be taking medications with similar mechanisms of action to duloxetine, but then those individuals could have been excluded (e.g. exclude people for medication use, not depression). 3. Page 9: Consider noting how large the randomization block sizes will be. 4. Although 60 mg is probably the optimal dose for neuropathic pain based on RCTs, in clinical trials examining the medication for knee OA, dosages up to 120 mg have been used (Chappell et al. 2009, 2011; Frankes et al. 2012). 5. Consider noting what rescue medications will be given. 6. Page 16: How will adverse events be assessed (open-ended interview, specific questions)?
--

VERSION 1 – AUTHOR RESPONSE

Reviewer: 1

- 1) This has been reflected in the abstract.
- 2) "Physiotherapy" removed.
- 3) Added.
- 4) This has been specified to KLG 2 or higher.

- 5) This has been clarified.
- 6) Clinical examination of the knee has been removed from experimental testing as it is only used during screening for confirmation of disease.
- 7) This has been reflected in the manuscript. References were included for scoring instructions.
- 8) Pain intensity measurement has been specified in the manuscript.
- 9) Response definition has been based on the randomised controlled trials by Chappel et al.
- 10) Patients are instructed to keep medication intake constant during the study intervention period; NSAIDS are allowed.
- 11) Outcomes and methods for quantification have been clarified.
- 12) Statistical methods have been clarified. Analysis will be per protocol.
- 13) Sample size calculation: this has been clarified.
- 14) Adherence is checked via telephone call during treatment and interview at the end of treatment period.

Reviewer: 2

1. Introduction has been shortened.
2. We agree with the reviewer that a study on the effects of duloxetine in OA pain patients with depression or anxiety would be interesting. This would however require doubling the sample size and challenging based on our previous experience using similar recruitment pathways. In similar cohorts (Alshuft et al, 2016) we found that OA patients with chronic pain had higher anxiety and lower mood scores but no clinically over depression. Hence, we decided to avoid uncontrollable outlier effects to limit recruitment to patients without clinically diagnosed depression or those on antidepressant medication.
3. Blocks of 2 and 9 were used. This has been specified in the manuscript.
4. We based our dosage on the fact that Chapell et al found no statistically significant differences between duloxetine 60 mg and duloxetine 120 mg groups in the MMRM analysis of the weekly 24-h average pain score or the 30% and 50% response rates at endpoint (Chappell, Amy S., et al. "Duloxetine, a centrally acting analgesic, in the treatment of patients with osteoarthritis knee pain: a 13-week, randomized, placebo-controlled trial." Pain 146.3 (2009): 253-260.)
5. Management of any adverse effects by the study team was limited to discontinuation of the study medication with further management of adverse effects or intolerable pain via standard NHS care pathways to which participants with adverse effects will be referred by clinical research fellow or clinical pharmacist.
6. Adverse events will be assessed by interviewing participants and referral to clinical care providers as appropriate.

VERSION 2 – REVIEW

REVIEWER	Leticia Deveza Rheumatology Department, Royal North Shore Hospital and Institute of Bone and Joint Research, Kolling Institute, University of Sydney, Australia.
REVIEW RETURNED	12-Jan-2017

GENERAL COMMENTS	Thank you for considering and addressing the comments and suggestions. I wish the authors success with the study.
---

REVIEWER	Steven Cohen Johns Hopkins, USA
REVIEW RETURNED	06-Jan-2017

GENERAL COMMENTS	This is an interesting study, but there is one nagging question I have. Compared to clinical trials designed to determine efficacy (pain relief as a primary outcome measure), this study is very small- only 27 patients in the "control" group. I realize the main objective is not to determine efficacy, but of course many people will interpret the results without considering this. This "lack" of power to detect a difference between groups is magnified even further because unlike for neuropathic pain (where 60 mg is similarly efficacious to 120 mg), studies evaluating duloxetine often titrated subjects to between 60 and 120 mg. Another minor point is that this study enrolls patient with knee OA, but the background (introduction) doesn't even mention the burden of knee OA (it discuss OA in general, but there are significant differences in functional capacity based on the joints affected).
---

VERSION 2 – AUTHOR RESPONSE

Thank you for the feedback. Regarding reviewer 2 points #1 and #2:

1 We understand the reviewer's concern from a clinical trials perspective. As argued in the manuscript of the study and appreciated by the reviewer, this is a mechanistic study and was powered on detecting mechanisms of the drug effect not clinical efficacy. Our planned sample size is significantly larger than similar published mechanistic studies using duloxetine (n=13 patients, 20 controls [Lopez-Sola et al., NPP 2010], 18 controls [Tendulkar et al., NPP 2011], 26 controls [Van Marle et al., NI 2011]) and also larger than the predictive study of duloxetine in pain (19 patients, 20 controls [Tetreault et al., PLoS Biology 2016]). Importantly, the clinical study results will be eligible* for data pooling including individual based meta-analysis and thereby in addition to the aims of the study may contribute to the wider knowledge base of clinical effects of duloxetine.

Needs checking with Research Ethics Committee

#2 We agree with the reviewer's suggestion and have included a statement on the burden of knee osteoarthritis, citing an estimation study from the Global Burden of Disease investigation from 2010 [Cross et al., 2013]. The statement was placed on the first paragraph of the "Background and rationale" section.

Additionally, we have revised the full manuscript for minor details such as spelling mistakes.